

# In vitro anticandidal activity and gas chromatography-mass spectrometry (GC-MS) screening of *Vitex agnus-castus* leaf extracts

Ibtisam Mohammed Ababutain[1,2] and Azzah Ibrahim Alghamdi[1,2]

[1] Basic & Applied Scientific Research Center (BASRC), Imam Abdulrahman Bin Faisal University, Dammam, Saudi Arabia
[2] Department of Biology, College of Science, Imam Abdulrahman Bin Faisal University, Dammam, Saudi Arabia

Corresponding author
Ibtisam Mohammed Ababutain,
iababutain@iau.edu.sa

## ABSTRACT

**Background:** Candida infections are becoming more drug resistant; it is necessary to search for alternative medications to treat them. Therefore, the present study estimates the anticandidal activity of *Vitex agnus-castus* (VA-C) leaf extracts.
**Methods:** We used the agar well diffusion method to assess the anticandidal activity of three different VA-C leaf extracts (ethanol, methanol, and water) against three *Candida* species (*Candida tropicalis*, *Candida albicans*, and *Candida ciferrii*). The minimum inhibitory concentration (MIC) was estimated using the two-fold dilution method and the minimum fungicidal concentration (MFC) was determined using the classic pour plate technique. The MFC/MIC ratio was calculated to estimate the microbicidal or microbiostatic activity. A gas chromatography mass spectrometer was used to screen the phytochemicals of the VA-C leaf extracts (ethanol, methanol, and water).
**Results:** All VA-C extracts ethanol, methanol, and water were significantly inhibited the growth of the test *Candida* species and the inhibition activity depended on the solvent used and the *Candida* species. The results showed that *C. tropicalis* was the most highly inhibited by all extracts followed by *C. albicans* and *C. ciferrii*. The MIC values were 12.5–25 µg/ml, and MFC values were 25–100 µg/ml. The ratios of MFC/MIC were two-fold to four-fold which was considered candidacidal activity. Ninety-five phytochemical compounds were identified by the GC-MS assay for the VA-C leaf extracts. The total number of compounds per extract differed. Methanol had 43 compounds, ethanol had 47 compounds, and water had 52 compounds. The highest compound concentrations were: 4,5-Dichloro-1,3-dioxolan-2-one in ethanol and methanol, 1H-Indene, 2,3-dihydro-1,1,2,3,3-pentamethyl in ethanol, Isobutyl 4-hydroxybenzoate in methanol, and Benzoic acid and 4-hydroxy- in water. These phytochemical compounds belong to different bioactive chemical group such as polyphenols, fatty acids, terpenes, terpenoids, steroids, aldehydes, alcohols, and esters, and most of which have anticandidal activity.
**Conclusions:** VA-C leaf extracts may be useful alternatives to anticandidal drugs, based on their effectiveness against all test *Candida* species at low concentrations. However, appropriate toxicology screening should be conducted before use.

# INTRODUCTION

The number of severe *Candida* infections is on the rise, which is concerning due to their virulence, ability to survive in extreme environments, and resistance to antifungal agents (*Paramythiotou et al., 2014*). *Candida* species can cause a wide variety of infections ranging from mild to severe, such as candidemia which has a mortality rate up to 38% in immunosuppressed patients (i.e., organ transplantation patients, patients under chemotherapy, HIV-infected, and diabetic) (*Koehler et al., 2019*; *De Oliveira Santos et al., 2018*). The rate of fungal infections, including candidiasis, can reach 20% in the intensive care unit and antifungal medications including azoles, echinocandins, fluoropyrimidines, and polyenes are typically used to treat these infections. However, determining the appropriate dose for treatment is challenging when considering their side effects (*Chatelon et al., 2019*). Candidiasis is one of the most common fungal diseases in the world and includes cutaneous candidiasis, mucosal candidiasis, onychomycosis, and systemic candidiasis. Healthy individuals are also susceptible to candidiasis (*De Oliveira Santos et al., 2018*). Genus *Candida* is deuteromycetes fungi and belongs to the Cryptococcaceae family, with up to 200 species. There are thirty species most commonly isolated in human infections including *Candida albicans*, *Candida tropicalis*, *Candida dubliniensis*, *Candida parapsilosis*, *Candida glabrata*, *Candida lusitaniae*, *Candida kefyr*, and *Candida krusei* (*Rodrigues, Rodrigues & Henriques, 2019*; *Kim, Jeon & Jae Kyung Kim, 2016*; *Brandt & Lockhart, 2012*; *Miceli, Diaz & Lee, 2011*).

Antifungals have a broad range of applications but it is difficult to determine the ideal treatment regime because their use can be limited and is often accompanied by side effects. The indiscriminate use of antibiotics has led to an increased resistance to these types of medications (*De Oliveira Santos et al., 2018*). Accordingly, researchers are exploring therapeutic alternatives, such as the use of plant essential oils or extracts. These have been proven beneficial in the treatment of several diseases due to their phytochemical components that have physiological and therapeutic effects on humans, limited toxicity, and low therapeutic costs (*Abdulrasheed et al., 2019*; *Sardi et al., 2013*). The World Health Organization reports indicate that up to 25% of modern medicines used in the United States of America originate from plants. In Africa and Asia, 80% of the population still uses medicinal herbs in their primary health care centers (*World Health Organization, 2002*). Moreover, there is documented evidence for the antimicrobial potential of more than 1,340 plants (*Yilar, Bayan & Onaran, 2016*). *Vitex* is one of the largest of the 250 genera in the family Verbenaceae found worldwide (*Ganapaty & Vidyadhar, 2005*). The therapeutic applications of *Vitex agnus-castus* (VA-C) and its safety as a medicinal plant are well stated (*Niroumand, Heydarpour & Farzaei, 2018*; *Neves & Da Camara, 2016*; *Rani & Sharma, 2013*). Previous studies have emphasized the antibacterial activity of the essential oils extracted from the seeds and fruit of VA-C (*Eryigit et al., 2015*; *Dervishi-Shengjergji et al., 2014*; *Ghannadi et al., 2012*). Other studies

have investigated the antimicrobial activity of essential oils extracted from the leaves of VA-C (*Katiraee et al., 2015*; *Ulukanli et al., 2015*). A few studies have demonstrated the antifungal activity of the seed oil (*Asdadi et al., 2014*). The antifungal potential of VA-C leaves essential oils against plant pathogens (*Yilar, Bayan & Onaran, 2016*). The antibacterial activity of the leaf extract of VA-C has been identified in a few studies (*Ababutain & Alghamdi, 2018*; *Kalhoro, Farheen & Aqsa, 2014*; *Arokiyaraj et al., 2009*) as well as the antimicrobial activity of VA-C leaf extract (*Kalhoro, Farheen & Aqsa, 2014*; *Maltaş et al., 2010*). These studies used only a few bacteria and one *Candida* species (*Candida albicans*). *Keikha et al. (2018)* evaluated the antifungal activity of ethanolic and aqueous lead extracts on *C. albicans* strains and they found that the ethanol extract was more effective than the aqueous extract against *C. albicans* strains. However, the effect of VA-C leaf extracts of against human *Candida* species has not been well-studied.

Therefore, this study aims to investigate the anticandidal activity and efficiency of VA-C leaf extracts (water, methanol and ethanol) against the three most frequently isolated *Candida* species (*Candida albicans*, *Candida tropicalis* and *Candida ciferrii*). We determined the phytochemicals of these extracts using Gas Chromatography-Mass Spectrometry (GC-MS).

## MATERIALS AND METHODS

### Plant material

*Vitex agnus-castus* VA-C leaves were collected from a private garden in Dammam City, Saudi Arabia belonging to Ibtisam Mohammed Ababutain. The plant was identified according to *Brickell & Zuk (1997)*.

### Preparation of plant extracts

Vitex agnus-castus leaves were washed with tap water and left to dry for 2 days at room temperature in a well-ventilated room using a fan to speed up the drying process, then ground to a fine powder. Maceration method described by *Pandey & Tripathi (2014)* was used with little modification, in which 60 g of the leaf powder was transferred to three Erlenmeyer flasks containing 300 ml of the three different solvents: distilled water, methanol (80%), and ethanol (80%) to a final concentration of 20% g/ml. The leaf mixtures were shaken for 72 h at 300 rpm/min/20 °C to extract the active compounds. We used the method previously described in *Ababutain (2019)* to extract the active compounds as follows: the leaf mixtures were filtered twice, first using Whatman No. 1 filter paper and then using bacterial filters. The filtrates were concentrated in an oven at 80 °C. The residues were re-suspended in dimethyl sulfoxide (DMSO) to a final concentration of 20%. All flasks were kept at 4 °C for further use.

### Agar well-diffusion method

Three different prepared VA-C leaf extracts with a 20% (mg/ml) concentration were screened for their anticandidal activity using the agar well-diffusion method (*National Committee for Clinical Laboratory Standards, 1993*) against three unicellular fungi. *Candida tropicalis* and *Candida albicans* were provided by King Fahd Hospital, Al Khobar,

Kingdom of Saudi Arabia. *Candida ciferrii* was obtained from the Biology Department, College of Science, Imam Abdulrahman Bin Faisal University.

Inoculums of the *Candida* species were prepared from new cultures in potato dextrose broth (PDB). A Biomerieux DensiCHEK plus meter device was used to adjust the cell suspension turbidity at $1–2 \times 10^6$ CFU/ml, which represents 0.5 McFarland standards. Each Petri dish was inoculated individually with 0.5 ml of the previous suspension. Melted potato dextrose agar (PDA) was poured over the inoculums, and the plates were rotated to ensure even distribution of the inoculums then left to harden at room temperate for 5 min. Five wells were made on the inoculated PDA using a 6 mm sterile cork-borer. Each well was filled with 100 µL of the plant extracts. Positive and negative controls were included; nystatin (10 mcg) was used as the positive control and DMSO was used as the negative control. The plates were incubated at 37 °C for 24 h. The anticandidal activity of the plant extracts was estimated in millimeters (mm) using a ruler and measuring the free growth zones around the wells. The experiments were performed in three replicates to ensure the reliability of the results.

## Determination of minimum inhibitory concentration

The minimum inhibitory concentration (MIC) of VA-C leaf extracts was estimated using the two-fold dilution method (*Omura et al., 1993*) as well as the method previously described in *Ababutain (2019)*. Briefly, the plant extracts were diluted with PDB media using 96-well microtiter plates in wells 1–10. Standard *Candida* inoculums at a concentration of $1–2 \times 10^6$ CFU/ml were transferred to the wells to make a final concentration of 50%. We used growth media with the *Candida* inoculum in well 11 and growth media with plant extracts in well 12, as positive and negative controls, respectively. The turbidity was examined by the naked eye after an overnight incubation period at 37 °C and the lowest concentration of plant extract showing no *Candida* species growth was recorded as MICs. All experiments were performed in three replicates.

## Determination of minimum fungicidal concentration

The classic pour plate technique was used to determine the minimum fungicidal concentration (MFC) (*National Committee for Clinical Laboratory Standards, 1997*). Concentrations that showed no *Candida* species growth from previous MIC experiments were transferred to Petri dishes, then 15 ml of melted PDA was poured over it and gently rotated and left to solidify. Inoculated Petri dishes were incubated at 37° for 48 h. The lowest concentration that showed no visible *Candida* species colonies were recorded as MFC (*Ababutain, 2019*). All experiments were performed in three replicates.

## Determination of anticandidal efficiency

The anticandidal efficiency of VA-C leaf extracts (ethanol, methanol and water) was determined by calculating the ratio of MFC/MIC according to *Levison & Levison (2009)*.

## Gas chromatography-mass spectrometry

We analyzed the bioactive compounds of all three VA-C leaf extracts (ethanol, methanol and water) with a gas chromatography-mass spectrometer (Shimadzu, Kyoto, Japan)

**Table 1 Anticandidal activity of VAC leaves extract at concentration of 20% by using well diffusion assay.**

| Candida species | Zone of inhibition (mm) Mean ± SD | | | | | |
|---|---|---|---|---|---|---|
| | Nystatin (10 mcg) | Negative control | Ethanol | Water | Methanol | P-value* |
| C. tropicalis | 11.0 ± 1.00 | 0 | 7.50 ± 0.50 | 5.67 ± 0.29 | 5.33 ± 0.29 | 0.01** |
| C. albicans | 5.83 ± 0.29 | 0 | 5.83 ± 0.29 | 5.00 ± 0.50 | 5.00 ± 0.50 | 0.047** |
| C. ciferrii | ND | 0 | 4.33 ± 0.58 | 3.33 ± 0.29 | 3.33 ± 0.29 | 0.037** |
| P-value | 0.01** | – | 0.01** | 0.01** | 0.01** | – |

Notes:
* P-value has been calculated using one-way ANOVA.
** Significant at $p < 0.01$ level. ND, not identified.

model QP2010 SE, with a 5 Sil MS 5% diphenyl/95% dimethyl polysiloxane capillary column (0.25-μm df, 30 meter, 0.25 mmID) using the method previously described in *Ababutain (2019)*. One microliter from each diluted plant extract (100/1,400, V/V in DMSO) was injected individually in the split mode with a split ratio of 1:10. We used the electron impact ionization system at 70 eV ionization energy to determine GC-MS exposure or detection. Pure helium (99.999%) was used as a carrier gas, at a constant column flow 0.7 ml/min and total flow of 10.4 ml/min. The flow control mode had a linear velocity of 29.6 cm/s. The injector temperature was set at 250 °C and the ion-source temperature was set at 250 °C. The column temperature was programed at 50–300 °C, with a hold time of 3 min, and a total run time of 29 min. The chemical compounds were identified using the National Institute of Standards and Technology (NIST 08) library match and the quantitative data were generated automatically as a percentage (*Adams, 2007*).

## Statistical analysis

The anticandidal activity of the VA-C leaf extract between the solvents and the *Candida* species was conducted using one-way ANOVA test. A *P*-value of <0.01 was considered statistically significant. Statistical data were analyzed using Statistical Packages for Software Sciences (*Statistical Packages for Software Sciences, 2013*) version 21 Armonk, New York, IBM Corporation.

# RESULTS

## Anticandidal activity of VA-C leaf extracts

The VA-C extracts were shown to inhibit the growth of all tested *Candida* species and the inhibition activity depended on the solvent type and *Candida* species. The results showed that *C. tropicalis* was the most inhibited by all the extracts followed by *C. albicans* and *C. ciferrii* (all *P* = 0.01). The effects of the ethanol extract against *C. tropicalis*, *C. albicans* and *C. ciferrii* were significantly higher compared to water and methanol extracts at *P* = 0.01, *P* = 0.037 and *P* = 0.047, respectively (Table 1).

Minimum inhibitory concentration results were between 12.5 μg/ml and 25 μg/mL and all extracts showed similar activity against all *Candida* species at MIC 25 μg/ml, except

**Table 2 Minimal Inhibitory Concentration (MIC) μg/ml and Minimal Fungicidal Concentration (MFC) μg/ml and their ratio of VA-C leaves extracts.**

| Candida species | Ethanol | | | Water | | | Methanol | | |
|---|---|---|---|---|---|---|---|---|---|
| | MIC | MFC | Ratio[*] | MIC | MFC | Ratio[*] | MIC | MFC | Ratio[*] |
| C. tropicalis | 12.5 | 25 | 2 | 25 | 50 | 2 | 25 | 50 | 2 |
| C. albicans | 25 | 50 | 2 | 25 | 50 | 2 | 25 | 100 | 4 |
| C. ciferrii | 25 | 50 | 2 | 25 | 50 | 2 | 25 | 50 | 2 |

Note:
[*] Ratio MFC/MIC.

*C. tropicalis* which was the most sensitive to the ethanol extract at MIC 12.5 μg/ml. The MFC results were between 25 μg/ml and 100 μg/ml. Most extracts showed similar MFC values against all *Candida* species at MFC 50 μg/mL except *C. tropicalis*. The MFC ethanol extract at 25 μg/ml had the highest anticandidal activity against *C. tropicalis* and the MFC methanol extract at 100 μg/ml was considered to be the lowest anticandidal activity against *C. albicans*. The results revealed that both MIC and MFC values for all three solvents were narrow where the differences between values were one to two concentrations only. The MFC/MIC ratio in all the three extracts were only two-fold to four-fold, which means that VA-C leaf extracts are potentially candidacidal (Table 2).

### Gas chromatography-mass spectrometry analysis

Our results revealed that VA-C leaf extracts are rich in phytochemical components of different concentrations. A total of 95 chemical compounds were extracted depending on the solvent type and, of these, 13 compounds were extracted by all three solvents and the total number of extracted compounds was 52 by water extraction, 47 by ethanol extraction, and 43 by methanol extraction (Table 3).

## DISCUSSION

Antibiotic resistance is becoming more common among a larger number of microorganisms, including *Candida* species, leading to a heightened interest in finding alternative treatments. The secondary metabolites of plants have made them useful for treating a variety of diseases, flavoring foods and products, preserving food, in pesticides, in perfumes and cosmetics, and more recently to inhibit the microbial growth. VA-C leaf extracts have been reported to cause mild and reversible side effects such as headache, acne, nausea, gastrointestinal disturbances, erythematous rash, pruritus, and menstrual disorders. However, no drug interactions have been associated with VA-C leaf extracts (*Daniele et al., 2005*). Therefore, VA-C leaf extracts (ethanol, methanol, and water) were investigated for their ability to inhibit the growth of three Azoles antibiotic-resistant *Candida* species: *C. ciferrii*, *C. albicans*, and *C. tropicalis* (*Romald et al., 2019*; *Bhakshu, Ratnam & Raju, 2016*).

Our results showed that alcohol extracts (methanol and ethanol) and aqueous extract have the ability to inhibit the growth of all tested *Candida* species. These results are in agreement with *Kalhoro, Farheen & Aqsa (2014)* study, which found that the ethanol

**Table 3 GC-MS analysis of VA-C leaves extracts, their molecular formula, nature and biological activities.**

| No | Compound name | Peak area % | | | Molecular formula | Compound nature and biological activities |
|----|---------------|---------|----------|-------|----------------|------------|
|    |               | Ethanol | Methanol | Water |                |            |
| 1 | 4,5-Dichloro-1,3-dioxolan-2-one | 7.43 | 7.45 | 1.39 | $C_3H_2Cl_2O_3$ | No report was found |
| 2 | Benzoic acid, 4-hydroxy- | 2.13 | 3.95 | 5.99 | $C_7H_6O_3$ | Phenolic compounds (*Eseyin et al., 2018*) |
| 3 | 5-Hydroxymethylfurfural | 1.18 | 1.61 | 0.82 | $C_6H_6O_3$ | Organic compound Antioxidant and Antiproliferative (*Ibrahim, Ali & Zage, 2016*) |
| 4 | Phenol | 1.12 | 1.73 | 1.41 | $C_6H_5OH$ | Phenolic compound, antiviral, antibacterial and antifungal activities (*Özçelik, Kartal & Orhan, 2011*) |
| 5 | 4H-Pyran-4-one, 2,3-dihydro-3,5-dihydroxy- | 0.75 | 0.82 | 1.41 | $C_6H_8O_4$ | Flavonoids, Anti-inflammatory, analgesic, antimicrobial activity (*Neeraj, Vasudeva & Sharma, 2019*) |
| 6 | Catechol | 0.50 | 0.57 | 1.14 | $C_6H_4(OH)_2$ | Polyhydric phenol, antiviral, antimicrobial activities (*Özçelik, Kartal & Orhan, 2011*) |
| 7 | Benzeneacetaldehyd, alpha-methyl- | 0.35 | 0.65 | 1.26 | $C_9H_{10}O$ | Hydrotropic aldehyde |
| 8 | Benzeneacetic acid, 4-hydroxy3-methoxy, | 0.39 | 0.38 | 0.58 | $C_{10}H_{12}O_4$ | No report was found |
| 9 | Pentanal | 0.10 | 0.16 | 0.88 | $C_5H_{10}O$ | alkyl aldehyde, Inhibition bacteria (*Lamba, 2007*) |
| 10 | Squalene | 0.13 | 0.37 | 0.40 | $C_{30}H_{50}$ | Terpenoid, Anticandidal activity, antioxidant, anti-inflammatory, and anticancer agent (*Ghimire et al., 2016*; *Zore et al., 2011*) |
| 11 | Maltol | 0.06 | 0.14 | 0.77 | $C_6H_6O_3$ | Antimicrobial activity (*Saud, Pokhrel & Yadav, 2019*) |
| 12 | 1H-Benzocyclohepten-7-ol,2,3,4,4a,5,6,7,8- | 0.36 | 0.12 | 0.11 | $C_{15}H_{26}O$ | Sesquiterpenids (*Solanki, Singh & Sood, 2018*) |
| 13 | n-Hexadecanoic acid | 0.70 | 0.56 | 0.96 | $C_{16}H_{32}O_2$ | Palmitic saturated Fatty acid ester, antimicrobial, antitumor activities, antioxidant, pesticide, nematicide, antiandrogenic and hypochloesterolemi (*Tyagi & Agarwal, 2017*; *Karthikeyan et al., 2014*; *Sermakkani & Thangapandian, 2012*;) |
| 14 | 3,5-Octadienoic acid, 7-hydroxy-2-methyl | 0.85 | 1.04 | – | $C_9H_{14}O_3$ | No report was found |
| 15 | Eugenol | 0.39 | 0.36 | – | $C_{10}H_{12}O_2$ | Phenolic compounds, antimicrobial activity, insecticide nematicide and food additive (*Tan & Nishida, 2012*; *Johny et al., 2010*) |
| 16 | 1,2,3-Benzenetriol | 0.56 | 0.67 | – | $C_6H_6O_3$ | No report was found. |
| 17 | Propylphosphonic acid, di(2-ethylhexyl) ester | 2.57 | 1.18 | – | $C_{21}H_{40}O_4$ | Ester |
| 18 | Methylparaben | 0.39 | 0.28 | – | $C_8H_8O_3$ | Antimicrobial activity, food preservative, added to cosmetic products, and pharmaceutical products (*Mincea et al., 2009*) |
| 19 | 1H-Cycloprop[e]azulen-7-ol, decahydro-1,1,7-trimethyl-4-methylene | 1.21 | 1.04 | – | $C_{15}H_{24}O$ | No report was found |
| 20 | Triacetin | 0.19 | 0.27 | – | $C_9H_{14}O_6$ | Triester of glycerin and acetic acid |
| 21 | 5-(1-Isopropenyl-4,5-dimethylbicyclo [4.3.0] | 1.13 | 0.87 | – | $C_{22}H_{36}O_2$ | No report was found |
| 22 | 2,4-Cholestadien-1-one | 1.72 | 0.96 | – | $C_{27}H_{42}O$ | No report was found |
| 23 | Phytol | 3.31 | 1.81 | – | $C_{20}H_{40}O$ | Diterpene, antiviral and antimicrobial activities (*Özçelik, Kartal & Orhan, 2011*) |
| 24 | 9,12,15-Octadecatrienoic acid, (Z, Z,Z)- | 1.18 | 0.90 | – | $C_{18}H_{30}O_2$ | Linolenic Omega-3 polyunsaturated fatty acid, anti–inflammatory (*Sermakkani & Thangapandian, 2012*) |

(Continued)

| No | Compound name | Peak area % | | | Molecular formula | Compound nature and biological activities |
|----|---------------|-------------|--|--|-------------------|-------------------------------------------|
| | | Ethanol | Methanol | Water | | |
| 25 | Cedran-diol, (8S,14)- | 0.13 | 0.52 | – | $C_{15}H_{26}O_2$ | No report was found |
| 26 | 3,7,11,15-Tetramethyl-2-hexadecen-1-ol | 1.11 | 1.34 | – | $C_{20}H_{40}O$ | Terpene Alcohol, antimicrobial, antioxidant, anti–inflammatory and flavoring agent (*Shibula & Velavan, 2015*; *Jegadeeswari et al., 2012*; *Sermakkani & Thangapandian, 2012*) |
| 27 | Spiro [4.5] dec-9-en-1-ol,1,6,6,10-tetramethyl | 0.54 | 0.39 | – | $C_{14}H_{24}O$ | No report was found |
| 28 | Dodeca-1,6-dien-12-ol, 6,10-dimethyl | 1.57 | 1.57 | – | $C_{14}H_{26}O$ | No report was found |
| 29 | Octadecanoic acid | 0.51 | – | 0.22 | $C_{17}H_{35}CO_2H$ | Stearic saturated fatty acid |
| 30 | Benzenediazonium, 2-hydroxy-, hydroxide, i | 0.86 | – | 1.17 | $C_6H_5N_2O$ | No report was found |
| 31 | Vitamin E | – | 0.39 | 0.36 | $C_{29}H_{50}O_2$ | Lipid, antibacterial, anti-alzheimer, antiaging and antioxidant (*Kumaravel, Muthukumaran & Shanmugapriya, 2017*; *Al-Marzoqi, Hadi & Hameed, 2016*; *Shahina et al., 2016*; *Al-Salih et al., 2013*) |
| 32 | Cedrol | – | 0.22 | 0.12 | $C_{15}H_{26}O$ | sesquiterpene alcohol |
| 33 | gamma-Sitosterol | – | 0.89 | 0.56 | $C_{29}H_{50}O$ | Steroid, antidiabetic drug (*Tripathi et al., 2013*) |
| 34 | Paromomycin | – | 0.12 | 0.22 | $C_{23}H_{45}N_5O_{14}$ | Treatment of diarrhea and protozoa infections (*Olajuyigbe et al., 2018*) |
| 35 | Heptanal | 0.20 | – | – | $C_7H_{14}O$ | aldehyde antibacterial activity (*Lamba, 2007*) |
| 36 | Ionone | 0.33 | – | – | $C_{13}H_{20}O$ | Sesquiterpenoids, antimicrobial agents (*Sharma et al., 2012*) |
| 37 | Chloroxylenol | 0.16 | – | – | $C_8H_9OCl$ | phenols with antiseptic activity, It is used in the manufacture of disinfectants and sterilizers (*McDonnell, 2009*) |
| 38 | 1-Heptadecene | 0.22 | – | – | $C_{17}H_{34}$ | unsaturated aliphatic hydrocarbons |
| 39 | Undecanal | 0.25 | – | – | $C_{10}H_{21}CHO$ | fatty aldehyde lipid molecule |
| 40 | 1H-Indene, 2,3-dihydro-1,1,2,3,3-pentamethyl | 9.63 | – | – | $C_{14}H_{20}$ | No report was found |
| 41 | Epiglobulol | 0.97 | – | – | $C_{15}H_{26}O$ | Alcohol |
| 42 | tau-Cadinol | 1.64 | – | – | $C_{15}H_{26}O$ | No report was found |
| 43 | alpha-Cadinol | 0.38 | – | – | $C_{15}H_{26}O$ | Antifungal activity (*Cheng et al., 2012*) |
| 44 | Phytol, acetate | 0.37 | – | – | $C_{22}H_{42}O_2$ | Food additive, antimicrobial, anti-inflammatory, anticancer and antidiuretic properties (*Sermakkani & Thangapandian, 2012*) |
| 45 | 1S,2S,5R-1,4,4-Trimethyltricyclo [6.3.1.0(2,5) | 1.26 | – | – | $C_{15}H_{24}$ | No report was found |
| 46 | beta-iso-Methyl ionone | 0.39 | – | – | $C_{14}H_{22}O$ | No report was found |
| 47 | Longipinane, (E)- | 0.41 | – | – | $C_{15}H_{24}$ | No report was found |
| 48 | (-)-Isolongifolol, methyl ether | 0.70 | – | – | $C_{16}H_{28}O$ | Ether |
| 49 | Taraxasterol | 0.07 | – | – | $C_{30}H_{50}O$ | Anti-tumor and chemopreventive activity (*Ovesná, Vachálková & Horváthová, 2004*) |
| 50 | S-Methyl methanethiosulphonate | 0.07 | – | – | $CH_3SO_2SCH_3$ | Ester, Antimutagenic agent and antimicrobial activity (*Joller et al., 2020*; *Miguel et al., 2016*) |
| 51 | 1-Heptatriacotanol | 0.23 | – | – | $C_{37}H_{76}O$ | Fatty alcohol |

| No | Compound name | Peak area % | | | Molecular formula | Compound nature and biological activities |
|---|---|---|---|---|---|---|
| | | Ethanol | Methanol | Water | | |
| 52 | 2-Vinylfuran | | 0.83 | | $C_6H_6O$ | Antimicrobial activity (*Drobnica & Sturdík, 1980*) |
| 53 | Salicyl hydrazide | – | 0.39 | – | $C_7H_8N_2O_2$ | Phenolic compounds, antimicrobial activity, Anti-inflammatory (*Madan & Levitt, 2014*) |
| 54 | Isobutyl 4-hydroxybenzoate | – | 8.91 | – | $C_{11}H_{14}O_3$ | No report was found |
| 55 | Methyl(ethenyl)bis(but-3-en-1-ynyl) silane | – | 1.62 | – | $C_7H_{16}Si$ | No report was found |
| 56 | beta Carotene | – | 0.16 | – | $C_{40}H_{56}$ | Carotenoids used as food, nutrition, antioxidant, disease control, and antimicrobial agents (*Kirti et al., 2014*) |
| 57 | 17-Norkaur-15-ene, 13-methyl-, (8.beta.,13.b | – | 1.05 | – | $C_{20}H_{32}$ | No report was found |
| 58 | 3-Hydroxy-2-(2-methylcyclohex-1-enyl) propan- | – | 1.48 | – | $C_{10}H_{16}O_2$ | No report was found |
| 59 | Cyclopropanebutanoic acid, 2-[[2-[[2-[(2-pen | – | 1.20 | – | $C_{11}H_{22}N_2O_4$ | No report was found |
| 60 | Cholan-24-oic acid, methyl ester, (5.beta.)- | – | 1.56 | – | $C_{25}H_{40}O_3$ | No report was found |
| 61 | Lup-20(29)-en-3-ol, acetate, (3.beta.)- | – | 1.12 | – | $C_{32}H_{52}O_2$ | No report was found |
| 62 | geranyl-.alpha.-terpinene | – | 0.80 | – | $C_{20}H_{32}$ | Terpinene |
| 63 | Tungsten, tricarbonyl-(2,5-norbornadiene) | – | – | 1.32 | $C_{14}H_{16}$ | No report was found |
| 64 | 1,2-Cyclopentanedione | – | – | 1.21 | $C_7H_{10}O_2$ | Prevents gastrointestinal tumor growth (*Neeraj, Vasudeva & Sharma, 2019*) |
| 65 | 2-Cyclopenten-1-one, 2-hydroxy-3-methyl- | – | – | 0.34 | $C_6H_8O_2$ | No report was found |
| 66 | 1,2,3-Propanetriol, 1-acetate | – | – | 1.23 | $C_5H_{10}O_4$ | No report was found |
| 67 | Acetoacetic acid, 3-thio-, benzyl ester | – | – | 0.16 | $C_{11}H_{12}O_2$ | No report was found |
| 68 | trans-Z-.alpha.-Bisabolene epoxide | – | – | 1.11 | $C_{15}H_{24}O$ | No report was found |
| 69 | 2-Hydroxyoctanoic acid | – | – | 0.62 | $C_8H_{16}O_3$ | No report was found |
| 70 | 1-Tetradecene | – | – | 0.67 | $C_{14}H_{28}$ | Antimicrobial activity (*Naragani et al., 2016*) |
| 71 | Benzoic acid, 4-methoxy- | – | – | 0.69 | $C_8H_8O_3$ | No report was found |
| 72 | Chlorozotocin | – | – | 0.20 | $C_9H_{16}ClN_3O_7$ | No report was found |
| 73 | 2-Isopropyl-5-methyl-6-oxabicyclo [3.1.0] hex | – | – | 1.51 | $C_{10}H_{16}O_2$ | No report was found |
| 74 | Quinic acid | – | – | 2.96 | $C_7H_{12}O_6$ | Anti-viral activity (*Özçelik, Kartal & Orhan, 2011*) |
| 75 | 3-Methylindene-2-carboxylic acid | – | – | 1.11 | $C_{11}H_{10}O_2$ | No report was found |
| 76 | O, O-Dibutyl S-(2-acetamidoethylmercapto) p | – | – | 1.32 | $C_{12}H_{22}O_4$ | No report was found |
| 77 | 3-Deoxy-d-mannonic acid | – | – | 1.21 | $C_6H_{12}O_6$ | No report was found |
| 78 | Cyclooctane-1,4-diol, cis | – | – | 0.44 | $C_8H_{16}O_2$ | No report was found |
| 79 | cis, cis, cis-7,10,13-Hexadecatrienal | – | – | 0.58 | $C_{16}H_{26}O$ | Unsaturated fatty aldehyde |
| 80 | 10-Iodo-7-oxa-2-thia-tricyclo [4.3.1.0(3,8)]de | – | – | 1.08 | $C_8H_{11}IOS$ | No report was found |
| 81 | Bicyclo [6.1.0] nonane, 9-(1-methylethylidene | – | – | 3.66 | $C_{12}H_{20}$ | No report was found |
| 82 | Inositol | – | – | 0.15 | $C_6H_{12}O_6$ | Essential nutrient, Cancer chemoprevention agent, treatment for Polycystic Ovary Syndrome and insulin sensitizing agent (*Carlomagno & Unfer, 2011*) |
| 83 | Xylose | – | – | 0.15 | $C_5H_{10}O_5$ | Pentose sugar (*Huntley & Patience, 2018*) |
| 84 | Scyllo-Inositol | – | – | 1.18 | $C_6H_{12}O_6$ | treatment of Alzheimer's disease (*Ma, Thomason & McLaurin, 2012*) |

(Continued)

| No | Compound name | Peak area % | | | Molecular formula | Compound nature and biological activities |
|----|---------------|---------|----------|-------|-------------------|-------------------------------------------|
| | | Ethanol | Methanol | Water | | |
| 85 | 2,4-Pentadien-1-ol, 3-pentyl-, (2Z)- | – | – | 0.94 | $C_{10}H_{18}O$ | No report was found |
| 86 | Widdrol hydroxyether | – | – | 0.23 | $C_{15}H_{26}O_2$ | No report was found |
| 87 | Stigmasterol | – | – | 0.21 | $C_{29}H_{48}O$ | Steroid, antioxidant, antimicrobial, anticancer, antiarthritic, antiasthma, anti-inflammatory, diuretic (*Tyagi & Agarwal, 2017*; *Kumar, Vasantha & Mohan, 2014*) |
| 88 | beta.-Amyrin | – | – | 0.43 | $C_{30}H_{50}O$ | Triterpenes, anti-inflammatory (*Okoye et al., 2014*) |
| 89 | 5,5′-Dihydroxy-3,3′-dimethyl-2,2′-binaphthal | – | – | 1.31 | $C_{17}H_{14}O_6$ | No report was found |
| 90 | Lanosterol | – | – | 0.22 | $C_{30}H_{50}O$ | Sterol, essential components of eukaryotic cells (*Wei, Yin & Welander, 2016*) |
| 91 | Betulin | – | – | 0.58 | $C_{30}H_{50}O_2$ | Anti-Viral and anti-tumour (*Tolstikov et al., 2005*) |
| 92 | alpha-Tocopheryl acetate | – | – | 0.23 | $C_{31}H_{52}O_3$ | Antimicrobial activity (*Bidossi et al., 2017*) |
| 93 | Geldanamycin | – | – | 0.33 | $C_{29}H_{40}N_2O_9$ | Chemotherapeutic agents (*Da Rocha, Lopes & Schwartsmann, 2001*) |
| 94 | Dihydrosteviobiside | – | – | 0.26 | C32H52O13 | No report was found |
| 95 | Bronopol | – | – | 0.10 | $C_3H_6BrNO_4$ | Antimicrobial activity (*Birkbeck et al., 2006*; *Treasurer, Cochrane & Grant, 2005*) |
| | Total compounds for each solvent | 47 | 43 | 52 | | |

VA-C leaf extract has the potential to inhibit the growth of *C. albicans*. Our results are also consistent with *Maltaş et al. (2010)* who observed that the methanol extract of VA-C leaves inhibits the growth of *C. albicans*. Moreover, we showed that the inhibitory capacity of the solvents varied significantly in descending order of ethanol, then water, then methanol. These results are in line with a recent study conducted by *Keikha et al. (2018)* who found that VA-C ethanol leaf extract has the highest inhibiting effect against *C. albicans* isolates than water extract. Our results showed a similarity in the inhibitory effect of all extracts with nystatin (10 mcg) as a positive control against *C. albicans*, where the inhibitory effect for the positive control was higher than all extracts against *C. tropicalis*.

We found that MIC showed that the ethanol extracts of VA-C were relatively higher than water and methanol. MIC values were between 12.5 µg/ml and 25 µg/ml for ethanol and represented a dilution of 4 and 3, respectively. For water and methanol the MIC values are specified at 25 µg/ml which represents dilution 3. Our results are similar to those of *Keikha et al. (2018)* who also found that the ethanol extract of VA-C was more effective than the aqueous extract when the MIC values of ethanol against isolates of *Candida* species were between 0.78 µg/ml and 1.56 µg/ml, which represent dilution 7 and 8, respectively. The values of the aqueous extract were between 6.25 µg/ml and 1.562 µg/ml, which represent dilutions of 5 and 7, respectively.

There was a convergence of MFC values, which represents only the three dilutions from 1 to 3 (100 µg/ml and 25 µg/ml), respectively. The VA-C extract of ethanol was the most

influential on *C. tropicalis* with the value of MFC 25 µg/ml and the aqueous extract was less effective on *C. albicans*, with a value of 100 µg/ml.

Selection of antibiotics for the treatment of infections is highly influenced by the mechanism of action. Antibiotics classified into either by killing the microbe (microbicidal) or inhibiting its growth (microbistatic) (*Etebu & Arikekpar, 2016*). Antibiotics with inhibitory effects are usually prescribed to patients who do not have problems with their immune system, while antibiotics with a fatal effect are prescribed for patients with low immunity or severe infections (*Davies & Davies, 2010*). *Candida* species are generally opportunistic and affect the group of people with low immunity so antibiotics that are prescribed are generally more effective if they are of the fatal type. Therefore, the inhibitory efficiency of the VA-C extract was estimate using the ratio between MFC and MIC. Our results showed that the ratio of MFC/MIC between two-fold to four-fold have a candidacidal effect (*Levison & Levison, 2009*). To our best of our knowledge, ours is the first study to establish this finding.

We found that the extracts of VA-C differed in their inhibitory effect according to the type of solvent and this is may be due to the difference in the degree of polarity between the solvent. Water has the highest polarity of 1,000 followed by methanol (0.762) and finally, ethanol (0.654). The compounds extracted by these highly polar solvents differ in quantity and quality (*Abubakar & Haque, 2020*). Many studies have demonstrated the effect of the solvent type on the inhibitory potential of plant extracts (*Aljuraifani, 2017*; *Ababutain, 2015*).

The GC-MC analysis result revealed that all three VA-C extracts were rich in chemical compounds that act as an anti-inflammatory, anticancer, anti-Alzheimer, anti-diarrheal, anti-diabetic, anti-viral, antioxidant, anti-allergic, nematicide, antibacterial, antifungal. These extracts are also used as food preservatives and flavorings, as previously found in other published works (Table 3). Several of these secondary metabolites belong to important chemical groups such as polyphenols, fatty acids, terpenes, terpenoid, steroids, aldehydes, alcohol, and esters. These results are in agreement with a previous study of *Keikha et al. (2018)*, which stated that the VA-C extract was rich in chemical compounds, and the alcoholic extract contained 36 chemical compounds that belong to different chemical groups. Our results showed that the majority of compounds were 4,5-Dichloro-1,3-dioxolan-2-one in both ethanol and methanol, 1H-Indene, 2,3-dihydro-1,1,2,3,3-pentamethyl in ethanol, Isobutyl 4-hydroxybenzoate in methanol, and Benzoic acid and 4-hydroxy- in water. *Keikha et al. (2018)* found that the majority of compounds in the VA-C ethanol extract were α-Pinene, isoterpinolene, caryophyllene, and azulene. The difference in the number of phytochemical compounds may be attributed to the variations among the VA-C genotypes (*Karaguzel & Girmen, 2009*).

The inhibitory activity of VA-C extracts maybe attributed to the presence of important bioactive compounds (*Abdal Sahib, Al-Shareefi & Hameed, 2019*), which may target different structures of the *Candida* species including the cell wall, cell membrane, and mitochondria enzymes. Some of these compounds may reduce or prevent the virulence factors, including adhesins, enzymes production, germ tubes (Pseudohyphal), biofilm formation, and quorum sensing (*De Oliveira Santos et al., 2018*; *Liu et al., 2017*;

*Sardi et al., 2013*). Our results showed the diversity of the compounds extracted from VA-C plant leaves that belong to several effective biochemical compounds with different anticandidal activity, including polyphenols that can destroy the *Candida* cell membrane leading to permeability of the cell contents (*Peralta et al., 2015*; *Hwang et al., 2011*; *Hwang et al., 2010*), inhibition of mitochondrial enzyme activity in the *Candida* cell (*Yang et al., 2014*) and inhibition of the germ tube formation (*Seleem et al., 2016*). Fatty acids with carbon chains between 10 and 12 carbons had a good inhibitory effect against *Candida* species (*Ababutain, 2019*; *Bergsson et al., 2001*). Terpenes have been reported to have inhibitory effects against *C. albicans* and may prevent biofilm formation (*Pemmaraju et al., 2013*). Terpenoids inhibit *C. albicans* cell growth by affecting the membrane and preventing adhesins, biofilm formation, and germ tube formation (*Touil et al., 2020*; *Raut et al., 2013*; *Zore et al., 2011*).

## CONCLUSIONS

Our results showed that VA-C leaf extract is rich in bioactive compounds with broad spectrum activity that inhibited all the tested *Candida* species despite different species. Accordingly, VA-C leaf extracts may inhibit the growth of *Candida* species in general, compared to antifungals that affect a specific species or a strain of species and require an accurate diagnosis of the *Candida* isolation to choose the appropriate antifungal. The inhibitory activity of the ethanol solvent was better than methanol and water, which may indicate the importance of choosing the appropriate solvent to extract phytochemicals with high inhibiting effectiveness and in higher quantities. Moreover, our results showed that the extract had a candidacidal effect on test *Candida* species at low concentrations, which may reduce the side effects of the extract. VA-C leaf extracts are advantageous, and a promising component that can be used to develop an alternative anticandidal agent. Further studies are required to assess the toxicity, genotoxicity and mutagenicity of VA-C extracts and prove their safety for human use.

## ACKNOWLEDGEMENTS

The authors thank the Director of Basic and Applied Scientific Research Centre (BASR) at Imam Abdulrahman Bin Faisal University, Dammam, Saudi Arabia for her continuous support and encouragement. The authors would like to thank Dr. Ahmed Alsayyah, Dr. Reem AlJindan, and Mrs. Nouf Alromaihi at King Fahd Hospital, Al Khobar, Kingdom of Saudi Arabia for providing us with the microorganisms for testing.

### Funding

The authors received no funding for this work.

### Competing Interests

The authors declare that they have no competing interests.

## Author Contributions

- Ibtisam Mohammed Ababutain conceived and designed the experiments, performed the experiments, analyzed the data, prepared figures and/or tables, authored or reviewed drafts of the paper, and approved the final draft.
- Azzah Ibrahim Alghamdi conceived and designed the experiments, performed the experiments, analyzed the data, prepared figures and/or tables, and approved the final draft.

## Data Availability

Raw data are available as Supplemental Files.

## Supplemental Information

Supplemental information for this article can be found online at http://dx.doi.org/10.7717/peerj.10561#supplemental-information.

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
