# Peer review of "In vitro anticandidal activity and gas chromatography-mass spectrometry (GC-MS) screening of *Vitex agnus-castus* leaf extracts"

_PeerJ, doi:10.7717/peerj.10561_

## Round 0.1 · original submission · Major Revisions

Dear Prof Ababutain,

Thank you for your submission to PeerJ. I sent your manuscript to reviewers and I have now received their reports which are copied below.

After careful evaluation of the manuscript and reviewers’ comments, I inform you that your manuscript can be reevaluated after major revisions. I ask you to pay special attention to the reports from Reviewers 1 and 3.

Reviewer 1 ·

Basic reporting

The MS “In vitro anticandidal activity and gas chromatography-mass spectrometry (GC-MS) screening of Vitex agnus-castus leaves extracts”, authored by Ibtisam Mohammed Ababutain and Azzah Alghamdi, describes the chemical characterization and the study of the anticandidal potential of the extracts extracted from leaves of Vitex agnus-castus plant. The protocol used in the chemical characterization is in agreement with literature procedures and the results obtained regarding the biological potential were quite promising. This work could be a relevant contribution in this direction and deserves publication in Peer J. However, there are some points to be addressed by the authors before a resubmission.

Experimental design

The research agrees with the Scope of the journal. The experimental part describes consolidated methodologies and promising results were obtained. However, in some cases the methods are not described with sufficient details or information to replicate. I suggest a detail review.

Validity of the findings

The results obtained are quite promising. However, they need to be better discussed. I suggest adding in the discussion the results obtained for the positive control indicating the potency and efficiency of the tested extracts. This comment applies to the results described in tables 1 and 2.

Additional comments

Major concerns:
#1. It’s important that in the Abstract other relevant information are included, besides those already presented, such as the majority compounds showed in the different extracts (ethanol, methanol and water).
#2. The leaves of plant were washed with tap water and dry for two days at room temperature (lines 102-103). Please, comment more about this. The ambiental conditions of room were control, like the humidity for example.
#3. Regarding the determination of the anticandidal efficiency of leaves extracts (line 150), how this was determined? Please, comment on this.
#4. The result and discussions are presents in a clear and objective way. However, I suggest adding in the discussion the results obtained for the positive control indicating the potency and efficiency of the tested extracts. This comment applies to the results described in tables 1 and 2.
#5. Regarding to lines 193-194 and 295-296, it was described that the extracts could be used as a promising source to develop an alternative anticandidal agent. Has any toxicological study been performed on this extract? Please add more details.

Minor points:
#1. Make a comparative study between the chemical compounds found in those extracts with other works already described in the literature. Other extracts are used to treat candida infections? Can you add this information on the introduction?
#2. Make a careful review for typos. For example, page 1 line 31 and page 2 line 50.

Reviewer 2 ·

Basic reporting

Text Corrections:

-Authors need to correct the words "cifrrii" for "ciferrii" in lines: 23 and 116.

-I would suggest to the authors to change or remove the sentence in lines 42-43, “…is possible to use this plant extracts in manufacturing of alternative anticandidal drugs.”
It is important to make sure that there are studies of toxicity, genotoxicity and mutagenicity of this plants to avoid adverse reaction.

-line 47; correct the sentence “…risk to human health. Due to their virulence, ability to survive…” for “…risk to human health, due to their virulence and ability to survive…”

-Table 1 – change de word “identfed” for “identified”

-Table 2 – only the first “ratio” written in the column has an “*”. Put on those two others also.

-Table 3 – In the third column, named as “Peak Area%”, authors must make clear what means “EL, ML, WL”, presumably refers to the extraction’s solvent.

Experimental design

- Authors have performed the determination of Minimum Fungicidal Concentration (MFC), which is an important and complementary result for Minimum Inhibitory Concentration (MIC).

- I miss in table 2 the use of a positive control, authors could use nystatin or fluconazole for example, as used by Oliveira et al.(http://dx.doi.org/10.1590/0001-3765201720170254)

- It is not clear in the section about Gas Chromatography-Mass Spectrometry (GC-MS), lines 155 to 168, that all 3 extracts were injected. Please provide more information.

- Authors claimed they calculated the ratio from MFC/MIC (Levison and Levison, 2009), the given reference says at page 792 that “…For bactericidal drugs, the MBC is usually the same as, and generally, not more than fourfold greater than, the MIC.”
The rations calculated by the authors (Table 2) seems to be not correct. Please recalculate them or provide better explanation.

Validity of the findings

The work has relevance once still lack of knowledge regarding VA-C against human Candida sp, as highlighted in lines 89-91. However, I am not convinced by the results and references provided that "... VA-C leaf extracts were superior to antibiotics in general, ..." as stated in the conclusions, lines 287-288. I suggest to remove this sentence.

Additional comments

Ababutain and Alghamdi, described the activity of Vitex agnus-castus (VA-C) leaves' extracts against three Candida species (Candida tropicalis, albicans and ciferrii), for that, they used three different extracts, from water, methanol and ethanol.

The work is relevant and well written. I can recommend this manuscript for publication if corrections are made, since the results obtained by the authors contribute to the knowledge of VA-C as a bioactive extract against the investigated pathogens.

Reviewer 3 ·

Basic reporting

 The manuscript is interesting since there is a worldwide trend in the study of natural extracts with biological activity, but as seen in the development of the text, there are many published works on the same topic, that is, this article is not unpublished.
 It should be clarified that candida is not yeast, it is a fungus.
 In materials and methods, I see that the concentration of the extracts was done in an oven and not in a rotary evaporator at low pressures, I think that extracts at 80 ° C can lose compounds or modify their structure, in the article by Amita Pandey, Shalini Tripathi (2014), the methodology is not indicated in the way that the authors of this article describe.
 Furthermore, with oven evaporation, the total evaporation of the solvents is not guaranteed, which leads to think that the antifungal action may be caused by the extract and residues of solvents such as ethanol.
 The way to express the concentration of the mixture of leaf powder and solvent is ambiguous and needs to be corrected.
 On line 111 is the number 33 that I don't make sense of.
 It is not written one μL is 1 μL.
 The list of References is very extensive since they placed the citations of table 3 of the “Compound nature and biological activities”.
 The English language should be improved to ensure that an international audience can clearly understand your text.

Experimental design

no comment

Validity of the findings

 In the results it is not known if the antifungal effect is due to the essential oil of the VA-C extracts leaves or the solvent used for the extraction since it was not completely eliminated.
 The discussion should be reviewed and summarized as information is repeated.
 It cannot be stated with the result of the GC-MC analysis that the extracts have significant activity such as anti-inflammatory, anticancer, anti-alzheimer, anti diarrhea, anti-diabetic, anti-viral, antioxidant, anti-allergic, nematicide, antibacterial, antifungal, as well as other uses as food, for this you have to do studies.
 In the text there is no information that leads me to Table 2, in addition this table lacks explanatory information.

Additional comments

Studying extracts with biological activity requires us to read several articles in order to design the methodology that we are going to develop in our research, we cannot use a method with which our results cannot be comparable.

Annotated reviews are not available for download in order to protect the identity of reviewers who chose to remain anonymous.

---

## Round 0.2 · accepted · Accept

Dear Dr Ababutain,

Thank you for sending the revised version of your manuscript.
As you can see, the reviewers were favorable to the publication of the new version of the manuscript.

I am pleased to inform you that your revised manuscript has been accepted for publication in PeerJ.

Reviewer 1 ·

Basic reporting

The MS “In vitro anticandidal activity and gas chromatography-mass spectrometry (GC-MS) screening of Vitex agnus-castus leaf extracts”, authored by Ibtisam Mohammed Ababutain and Azzah Alghamdi, describes the chemical characterization and the study of the anticandidal potential of the extracts extracted from leaves of Vitex agnus-castus plant. The protocol used in the chemical characterization agrees with literature procedures and the results obtained regarding the biological potential were quite promising. This work could be a relevant contribution in this direction and deserves publication in Peer J. The authors made the necessary adaptions suggested in the previous review.

Experimental design

The research agrees with the Scope of the journal and the experimental part describes consolidated methodologies and promising results were obtained.

Validity of the findings

The results obtained are quite promising and after the changes made to the text, the results became clearer and more objective.

Additional comments

The authors made the necessary adjustments in the text and I recommend the publication of this MS in the current form.

Reviewer 2 ·

Basic reporting

Authors properly performed all the given suggestions.

Experimental design

Authors properly performed all the given suggestions and clarified doubts.

Validity of the findings

Corrections were made.

Additional comments

As I stated previously, the work of Ababutain and Alghamdi is relevant.

Since all correction were applied and some doubts were clarified, I recommend this work for publication on PeerJ.